# Contribution of TGF-β1 and Effects of Gene Silencer Pyrrole-Imidazole Polyamides Targeting TGF-β1 in Diabetic Nephropathy

**DOI:** 10.3390/molecules25040950

**Published:** 2020-02-20

**Authors:** Shu Horikoshi, Noboru Fukuda, Akiko Tsunemi, Makiyo Okamura, Masari Otsuki, Morito Endo, Masanori Abe

**Affiliations:** 1Division of Nephrology, Hypertension and Endocrinology, Department of Medicine, Nihon University School of Medicine, Tokyo 173-8610, Japan; horiko19830823@yahoo.co.jp (S.H.); tsunemi.akiko@nihon-u.ac.jp (A.T.); makiyo91supika@yahoo.co.jp (M.O.); mi.piace.hayden.v@gmail.com (M.O.); 2Nihon University Research Center, Tokyo 173-8610, Japan; 3Faculty of Human Health Science, Hachinohe Gakuin University, Hachinohe, Aomori 031-8588, Japan; mendo@hachinohe-u.ac.jp

**Keywords:** diabetic nephropathy, TGF-β1, pyrrole-imidazole polyamide, rat, podocyte

## Abstract

TGF-β1 has been known to induce diabetic nephropathy with renal fibrosis and glomerulosclerosis. DNA-recognized peptide compound pyrrole-imidazole (PI) polyamides as novel biomedicines can strongly bind promoter lesions of target genes to inhibit its transcription. We have developed PI polyamide targeting TGF-β1 for progressive renal diseases. In the present study, we evaluated the contribution of TGF-β1 in the pathogenesis of diabetic nephropathy, and examined the effects of PI polyamide targeting TGF-β1 on the progression of diabetic nephropathy in rats. For in vitro experiments, rat renal mesangial cells were incubated with a high (25 mM) glucose concentration. Diabetic nephropathy was established in vivo in eight-week-old Wistar rats by intravenously administering 60 mg/kg streptozotocin (STZ). We examined the effects of PI polyamide targeting TGF-β1 on phenotype and the growth of mesangial cells, in vitro, and the pathogenesis of diabetic nephropathy in vivo. High glucose significantly increased expression of TGF-β1 mRNA, changed the phenotype to synthetic, and increased growth of mesangial cells. STZ diabetic rats showed increases in urinary excretions of protein and albumin, glomerular and interstitial degenerations, and podocyte injury. Treatment with PI polyamide targeting TGF-β1 twice weekly for three months improved the glomerular and interstitial degenerations by histological evaluation. Treatment with PI polyamide improved podocyte injury by electron microscopy evaluation. These findings suggest that TGF-β1 may be a pivotal factor in the progression of diabetic nephropathy, and PI polyamide targeting TGF-β1 as a practical medicine may improve nephropathy.

## 1. Introduction

Diabetic nephropathy is one microangiopathic complication of diabetes mellitus and the greatest cause of end-stage renal failure, in which 16,000 patients newly begin dialysis therapy every year in Japan. The development of radical medicines for diabetic nephropathy is an urgent matter to prevent an increasing number of patients with end-stage renal failure.

The early stage of diabetes mellitus induces thickening of the basement membrane, podocyte injury by apoptosis and functional abnormality, and endothelial damage in the glomerulus, arising from the loss of glomerular barrier function. Diabetes mellitus also induces mesangial dysfunction, obstruction of the capillary arteries and nodular lesions that induce glomerular sclerosis [1].

TGF-β comprises a large family of cytokines that help regulate growth, differentiation and morphogenesis in many cell types [2]. TGF-β has been found to be a critical factor in renal diseases such as glomerulosclerosis and mesangioproliferative glomerulonephritis. It stimulates mesangial cell (MC) proliferation and the production of extracellular matrix (ECM) components and induces epithelial-mesenchymal transformation (EMT) in renal tissue, which is critical in the pathogenesis of renal injury [3]. High glucose levels in diabetes mellitus induce TGF-β1 gene expression and the synthesis of renal ECMs that cause glomerular sclerosis [4]. Increases in TGF-β1 induce endothelial-mesenchymal transition (EndMT) to suppress glomerular endothelial function resulting from a failure of crosstalk between glomerular endothelial cells and podocytes, and then podocyte injury associates proteinuria [5]. Thus, TGF-β1 is a main molecule in the pathogenesis of diabetic nephropathy.

Pyrrole-imidazole (PI) polyamides are peptide compounds recognized by DNA that were first identified from antibiotics, such as duocarmycin A and distamycin A. As aromatic rings of the amino acids N-methylpyrrole and N-methylimidazole, PI polyamides recognize and bind to DNA according to specific sequences [6,7]. They can form hydrogen bonds with high affinity and specificity to double-stranded DNA (dsDNA), which is stronger than the bonds formed between protein and dsDNA. PI polyamides also inhibit protein interactions and some DNA transcription factors [8]. dsDNA recognition is based on side-by-side pairing of pyrrole (Py) with imidazole (Im) in the minor groove. Im paired opposite to Py targets the G-C base pair, whereas the opposite pairing (Py-Im) targets the C-G base pair. The pairing of Py-Py targets A-T and T-A base pairs [7]. PI polyamides are able to fully resist nuclease-induced biological degradation and do not require vector-assisted delivery systems because they can easily permeate cells and enter the nuclei. Thus, as novel DNA-recognized agents, PI polyamides may be more applicable than nucleic acid medicines [9]. Synthetic PI polyamides, designed to target gene promoters, might be effective as practical medicines that can regulate gene transcription.

We showed significant inhibition in human cells of TGF-β1 promoter activity and the expression of TGF-β1 mRNA and protein by PI polyamides, targeting the human TGF-β1 promoter [10]. We further showed that a PI polyamide targeting TGF-β1 could effectively retard the progression of renal diseases [11,12], post-angioplasty carotid artery stenosis [13], cornea fibrosis by alkali burns [14], skin hypertrophic scars [15], liver fibrosis [16], and encapsulating peritoneal sclerosis [17] in rats.

In the present study, we evaluated the contribution of TGF-β1 and examined the effects of PI polyamide targeting TGF-β1 in the pathogenesis of diabetic nephropathy in rats.

## 2. Results

### 2.1. Expression of TGF-β1 and Phenotype Marker mRNAs in MCs

Figure 1 shows the time course of the expression of TGF-β1 and phenotype marker mRNAs in MCs before, and after, changes from medium to high glucose concentrations. The abundance of TGF-β1 mRNA increased significantly (*p* < 0.05) at 6 and 9 h after changes from medium to high glucose. Increases in the abundance of TGF-β1 mRNA were transient and peaked at 6 h (Figure 1A). The abundance of osteopontin mRNAs was significantly (*p* < 0.05) increased at 6, 9, and 12 h after changes from medium to high glucose (Figure 1B). The abundance of αSMA mRNAs decreased significantly (*p* < 0.05) at 6, 9 and 12 h after changes from medium to high glucose (Figure 1C). These results indicate that a high glucose condition increased the expression of TGF-β1 mRNA and induced the synthetic phenotype of MCs.

### 2.2. Effects of TGF-β1 PI Polyamide on Expression of TGF-β1 and Osteopontin mRNAs, and Growth of MCs

A dose of 0.01 µM TGF-β1 PI polyamide significantly (*p* < 0.05) decreased the abundance of TGF-β1 mRNA in MCs with high glucose (Figure 2A). However, TGF- β1 PI polyamide did not affect the abundance of osteopontin mRNA in MCs with high glucose (Figure 2B). Figure 2C shows the effects of TGF-β1 PI polyamide on the growth of MCs with high glucose by WST-1 assay. High glucose significantly (*p* < 0.05) increased, and TGF-β1 PI polyamide significantly (*p* < 0.05) decreased, cell viability of the MCs with high glucose. Mismatch PI polyamide did not affect cell viability (Figure 2C).

### 2.3. Morphology of the Glomerulus in STZ Diabetic Rats

Figure 3 shows the morphological changes in the glomerulus in diabetic rats at 3 months after the injection of STZ. Hematoxylin and eosin staining showed enlargement of the glomerulus with mesangial proliferation. The diameter of glomerulus was 42.4 ± 4.3 μm (*n* = 15) in Control rats and 49.2 ± 4.5 μm (*n* = 15) in STZ diabetic rats. The diameter of glomerulus in STZ diabetic rats was significantly (*p* < 0.05) larger than in Control rats. Immunofluorescence of podocin shows lower staining of podocin and its interruption of the continuity of the glomerulus in diabetic rats at 3 months after injection of STZ compared to Control rats.

### 2.4. Delivery of PI Polyamide in Rat Kidney

Figure 4 shows the delivery of 2.5 mg/body of fluorescein isothiocyanate (FITC)-labeled TGF-β1 PI polyamide by intraperitoneal injection into rat kidney. PI polyamide was mainly distributed into the nucleus of the nephron tubule, but was slightly distributed into the glomerulus, at 1 day after injection. PI polyamide was distributed into the nucleus of the nephron tubule and glomerulus at 3 days after injection. PI polyamide had mostly disappeared from the nephron tubule and glomerulus at 7 days after the injection.

### 2.5. Serological Parameters in Peripheral Blood

Table 1 shows the serological parameters of peripheral blood from Control rats and STZ diabetic rats without or with Mismatch PI polyamide or TGF-β1 PI polyamide. Treatments with Mismatch PI polyamide and TGF-β1 PI polyamide for 3 months did not affect blood levels of glucose, hemoglobin A1c (HbA1c), blood urea nitrogen (BUN), and creatinine.

### 2.6. Effects of TGF-β1 PI Polyamide on Urinary Excretions of Protein and Albumin in STZ Diabetic Rats

Urinary excretion of protein were significantly higher in STZ diabetic rats than those in the Control rats at 9 (*p* < 0.05), 10, 11 and 12 (*p* < 0.01) weeks after STZ injection (Figure 5A). Urinary excretions of albumin (*p* < 0.05) were significantly higher in STZ diabetic rats than those in Control rats at 11 and 12 weeks after STZ injection (Figure 5B). Treatment with TGF-β1 PI polyamide decreased urinary excretions of protein in STZ diabetic rats at 11 and 12 weeks, compared to treatment with Mismatch PI polyamide, but statistically not significant (Figure 5A). Treatment with TGF-β1 PI polyamide did not significantly decrease urinary excretion albumin in STZ diabetic rats compared to treatment with Mismatch PI polyamide (Figure 5B).

### 2.7. Effects of TGF-β1 PI Polyamide on Expression of TGF-β1 Protein in Kidney from STZ Diabetic Rats

Figure 6 shows Western blot analysis for the expression of TGF-β1 protein in kidneys from STZ diabetic rats. The abundance of TGF-β1 protein was significantly (*p* < 0.05) higher in kidneys from STZ diabetic rats with Mismatch PI polyamide than that from Control rats. Treatment with TGF-β1 PI polyamide significantly (*p* < 0.01) decreased the abundance of TGF-β1 protein in kidneys from STZ diabetic rats.

### 2.8. Effects of TGF-β1 PI Polyamide on Renal Injury in STZ Diabetic Rats

Figure 7 shows a comparison of the glomerular injury score (GIS) and tubulointerstitial injury score (TIS) of kidneys from control rats and STZ diabetic rats treated with Mismatch PI polyamide or TGF-β1 PI polyamide at 3 months after STZ injection. Treatment with TGF-β1 PI polyamide significantly (*p* < 0.05) decreased the GIS (Figure 7B) and TIS (Figure 7C) of kidneys compared to the Mismatch PI polyamide.

### 2.9. Electron Microscopy Findings of Podocytes in STZ Diabetic Rats

Figure 8 shows electron microscopic findings of kidneys from Control rats, STZ diabetic rats and STZ diabetic rats treated with TGF-β1 PI polyamide at 3 months after STZ injection. Compared to the podocytes of the kidney from Control rats, fusion of the foot process and disappearance of the slit structure of the glomerulus are evident in the STZ diabetic rats. Treatment with TGF-β1 PI polyamide impaired the levels of the fusion of the foot process and the disappearance of the slit structure of the glomerulus.

## 3. Discussion

In regard to the mechanisms of the increase of TGF-β1 in diabetes mellitus, high glucose is known to activate protein kinase C and polyol pathway flux to increase radical oxygen species and advanced glycogen end-product [18]. These conditions stimulate transcription factors USF1/2, AP-1 and CREB that activate several genes to accumulate ECM, and which induce inflammation and glomerulosclerosis in the kidney [19]. Exposure to high glucose and fatty acid decreases AMP-kinase induces the activation of NFkB and the translocation of USF1 into the nucleus, which stimulates the transcription of TGF-β1 in the hyperglycemic condition [20]. The increased TGF-β1 is involved in the pathogenesis of diabetic nephropathy.

Mesenchymal cells, including renal MCs, show a contractile or synthetic phenotype. MCs in a normal glomerulus show the contractile phenotype, which do not proliferate, whereas MCs in an abnormal glomerulus by diabetes mellitus or hypertension shows the synthetic phenotype, inducing hyperproliferation with increases in organelles to produce many cytokines [21]. Increases in osteopontin, one of the synthetic phenotype markers and an inflammatory cytokine, strongly correlate with urinary albumin excretion and glomerulosclerosis in diabetic nephropathy [22]. The increased TGF-β1 contributes to the formation of glomerular ECM and albuminemia, seen with podocyte injury. In addition, increased USF1 in diabetes mellitus also binds to the promoter of osteopontin, which induces mesangial proliferation that leads to glomerular hypertrophy.

In the present study, the expression of TGF-β1 was transiently increased, the expression of osteopontin was also increased, and the expression of αSMA as a contractile phenotype marker was decreased after culture in the high glucose medium. In addition, the high glucose condition significantly stimulated the growth of MCs, whereas TGF-β1 PI polyamide significantly decreased the growth of MCs under this condition. These results indicate that the high glucose condition increased TGF-β1 and osteopontin, which contribute to the synthetic phenotype and the increased growth of MCs in culture.

We first elucidated the delivery of PI polyamide into the kidney after the intraperitoneal injection. FITC-labeled TGF-β1 PI polyamide was distributed into the nucleus of the nephron tubule and glomerulus at day 3, and had mostly disappeared from the tissues by day 7, after intraperitoneal injection. On the basis of these results, we injected TGF-β1 PI polyamide twice a week, for 3 months into the STZ diabetic rats. Previously, we investigated the pharmacokinetics of PI polyamides injected into rats. We found that the FITC-labeled PI polyamide strongly localized in the nuclei of the nephron tubule and glomerulus, and in the nuclei of mid-layer smooth muscle cells in the aorta, lung, and liver, with no other drug delivery system needed [11]. Thus, PI polyamides can distribute into the organs and strongly bind to their nuclei by themselves. This delivery property of PI polyamide administered into the body could be a potential advantage as medicines, compared to that of nucleic acid medicines. In addition, the PI polyamides were excreted mainly into the urine, and low molecular weight PI polyamide was excreted partially into the bile [23]. The property of urinary excretion of PI polyamide could also be advantageous in progressive renal diseases.

The intraperitoneal administration of TGF-β1 PI polyamide inhibited the expression of TGF-β1, and ameliorated the degeneration of nephron tubules and glomeruli in kidney of STZ diabetic rats. These results indicate that TGF-β1 PI polyamide could be effective in diabetic nephropathy with glomerular mesangial proliferation and interstitial fibrosis. In addition, kidneys from STZ diabetic rats showed an interruption in the continuity of podocytes in the glomerulus. Electron microscopy showed fusion of the foot process and disappearance of the slit structure of glomerulus from the STZ diabetic rats, which improved with the administration of TGF-β1 PI polyamide. These results indicate that increased TGF-β1 is involved in podocyte injury. Recently, crosstalk between glomerular endothelial cells and podocytes was established [24]. Podocytes generate many molecules including cytokines such as VEGF, CXCR4 and Tie-2. These molecules act to maintain the function of glomerular endothelial cells. However, these cells produce activated protein C to maintain podocyte function [25]. TGF-β1 is known to induce EndMT by which endothelial function is suppressed by the mesenchymal transformation [26]. The increased TGF-β1 is thought to contribute to impaired functions of glomerular endothelial cells and podocytes by disrupting their crosstalk in diabetes mellitus. Thus, treatment with TGF-β1 PI polyamide might be effective in improving the injury caused by diabetes mellitus by suppressing podocyte injury via improvement of EndMT, as well as the crosstalk between glomerular endothelial cells and podocytes that occur with this disease.

We examined the effects of treatments with TGF-β1 PI polyamide on STZ-induced diabetic nephropathy and found that TGF-β1 PI polyamide effectively improved diabetic nephropathy including degeneration of the nephron tubules and glomeruli, and podocyte injury. Thus, it is feasible that TGF-β1 PI polyamide may prevent the progression of human diabetic nephropathy. We recently developed a PI polyamide to target human TGF-β1 and examined this PI polyamide in a primate model of cyclosporine A-induced nephropathy and unilateral urethral obstruction in the common marmoset, which has high gene homology of 90% to the human TGF-β1 gene. The human TGF-β1 PI polyamide effectively improved these nephropathies in the common marmoset [27]. We will, therefore, create diabetic mellitus in the common marmoset with STZ, and examine the effectiveness of human TGF-β1 PI polyamide as a preclinical study for the treatment of diabetic nephropathy.

In conclusion, we found mesangial proliferation with change in the synthetic phenotype and podocyte injury, which were improved with TGF-β1 PI polyamide in the present experiments. These findings suggest that TGF-β1 could be a pivotal factor in the progression of diabetic nephropathy, and as a practical medicine, TGF-β1 PI polyamide could feasibly improve diabetic nephropathy.

## 4. Materials and Methods

### 4.1. Ethics

This investigation conformed to the Guide for the Care and Use of Laboratory Animals published by the US National Institutes of Health (NIH Publication No. 85-23, 1996). The ethics committee of the Nihon University School of Medicine approved the research protocols involving the use of living animals in this study (approval no. 11-034).

### 4.2. Culture of MCs and Glucose Stimulation

Glomeruli were isolated from the kidneys of 8-week-old male Wistar Kyoto/Izm rats (Japan SLC, Hamamatsu, Japan) with a graded-sieve technique described previously [28]. The renal cortex from three rats in each strain were excised and minced into small pieces under sterile conditions. After being pressed through a 200-mm sieve, the minced cortex was suspended in RPMI 1640 medium (GIBCO Laboratories, Grand Island, NY, USA). This suspension was then passed through a 120-mm sieve, and glomeruli collecting on the surface of the sieve were gathered and resuspended in RPMI1640. MCs were then isolated from explants of whole glomeruli according to the differential growth capacities of the glomerular epithelial cells and MCs. We confirmed that no cellular cytotoxicity and no cytolysis had occurred by measuring lactate dehydrogenase released into the culture medium of the MCs (Thermo Fisher Scientific, Waltham, MA, USA). MCs were then grown in DMEM containing 10% calf serum for 24 h, following which the culture medium was changed to DMEM with 0.2% FBS. Quiescence was established by incubating the cells in this medium for 48. The MCs were first cultured with normal glucose medium (glucose 5.6 mM [1000 mg/L] in DMEM) by 80% confluency and then with high glucose medium (glucose 25 mM [4500 mg/L] in DMEM).

### 4.3. RNA Extraction and Real-time PCR Analysis

A TRIzol reagent (Invitrogen, Carlsbad, CA, USA) was used to extract total RNA from the MCs. Total RNA (1 μg) was reverse transcribed into cDNA with random 9-mers with a Takara RNA PCR Kit (AMV) version 3.0 (Takara Bio, Ohtsu, Japan). Assay-on-Demand primers and probes (TGF-β1 [Rn00572010_m1], Acta2 [Rn01759928_g1], and Spp1 [Rn00681031_m1]) and osteopontin were obtained from Applied Biosystems Life Technologies (Tokyo, Japan). The mRNA was quantified with an ABI Prism 7300 (Applied Biosystems, Waltham, MA, USA). Each sample, with each reaction comprising 5 μL complementary DNA (total volume, 25 μL), was run in triplicate. We used glyceraldehyde-3-phosphate dehydrogenase (GAPDH) (4351317; Life Technologies, Carlsbad, CA, USA) with TaqMan Ribosomal RNA Control Reagents (Applied Biosystems) to control sample loading. Amplification conditions were 50 °C for 2 min, 95 °C for 10 min, 60 cycles of denaturation (95 °C for 15 s), and combined annealing-extension at 60 °C for 1 min. After we determined the threshold cycle (Ct), we used the comparative Ct method to calculate the relative quantification of the marker gene mRNA expression.

### 4.4. Synthesis of PI Polyamides Targeting Rat TGF-β1

The chemical structure of the PI polyamide targeting rat TGF-β1 promoter is shown in Figure 9. To obtain specificity to the rat TGF-β1, we designed the polyamide targeting rat TGF-β1 to span the boundary of the AP-1 binding site (22303 to 22297) of the TGF-β1 promoter, as described previously [11]. The mismatch polyamide was designed not to bind to the transcription sites of the promoter.

To create the PI polyamides by machine-assisted automatic synthesis of hairpin-type PI polyamides, we induced substitution of Im and Py with a PSSM-8 continuous-flow peptide synthesizer (Shimazu, Kyoto, Japan) at 0.1-µmol scale (200 mg of Fmoc-b-alanine-CLEAR Acid Resin, 0.50 mEq/g; Peptide Institute, Osaka, Japan). To perform the automatic solid-phase synthesis, we performed the first wash with dimethylformamide (DMF), then removed the Fmoc group with 20% piperidine/DMF, washed with methanol, coupling with a monomer for 60 min in an environment of 1-[bis(dimethylamino)methylene]-5-chloro-1H-benzotriazolium 3-oxide hexafluorophosphate and diisopropylethylamine (4 eq each), washed again with methanol, protected with acetic anhydride/pyridine, and performed a final wash with DMF. After the Fmoc group was removed from the Fmoc-β-alanine-Wang resin, it was successively washed with methanol. Fmoc-amino acid was used to perform the coupling step, followed by an additional methanol wash. These steps were repeated until all sequencing was complete. Thereafter, we protected and washed the N-terminal amino group with DMF and drained the reaction vessel. Next, following the cleavage step (5 mL of 91% trifluoroacetic acid-3% triisopropylsilane-3% 5 dimethylsulfide-3% water/0.1 mM resin), we performed cold ethyl ether precipitation to isolate the synthetic polyamides. After a further cleavage step (5 mL of N, N-dimethylaminopropylamine/0.1 mM resin, 50 °C overnight), we isolated the synthetic polyamides again by cold ethyl ether precipitation. Finally, we purified the polyamides by high-performance liquid chromatography (HPLC) with use of a PU-980 HPLC pump, UV-975 HPLC UV/VIS detector (Jasco, Easton, MD, USA), and a Chemcobound 5-ODS-H column (Chemco Scientific, Osaka, Japan).

### 4.5. Evaluation of MC Growth

Rat MCs were plated in 96 well dishes at 5 × 10^3^ cells/well with 10% fetal bovine serum (FBS) in normal glucose DMEM at 37 °C in 5% CO₂ for 24 h, after which the culture medium was changed to DMEM with 0.2% FBS for 24 h to establish quiescence. The cells were then stimulated with 10% FBS in high-glucose DMEM with 0.01 μM PI polyamide to TGF-β1 or Mismatch polyamide for 24 h. Premix WST-1 (TAKARA Bio, Shiga, Japan) at 10 μL/well was added to the cells for 90 min. The absorbance of the samples was read with a microtiter plate reader against a background control as a blank. The wavelength used to measure the absorbance of the formazan product ranged between 420 to 480 nm.

### 4.6. Distribution of FITC-labeled PI Polyamide in Kidney

PI polyamide targeting TGF-β1 was labeled with FITC (Sigma-Aldrich, St. Louis, MO, USA). To evaluate the distribution of PI polyamide in kidney, we intraperitoneally injected 1 mg of FITC-PI polyamide into Wistar rats. After 1, 3 and 7 days, we removed the kidneys from the rats, prepared frozen specimens, and viewed them.

### 4.7. Creation of Diabetic Rats and Treatment with PI Polyamide

The experimental protocol used to create the diabetic rats and administrations of Mismatch or TGF-β1 PI polyamide are shown in Figure 10. Male Wistar rats (Japan Charles River Laboratories, Yokohama, Japan) weighing 200 to 250 g were allowed free access to water and regular laboratory chow. Diabetes was induced in the rats by a single intraperitoneal injection of streptozotocin (STZ) (Sigma-Aldrich, 60 mg/kg body weight) according to a previous report [29]. Blood glucose was measured from tail vein blood using the *o*-toluidine method in the non-fasting conditions. Rats whose blood glucose level was at least 300 mg/dL 24 h after STZ treatment were subjected to further study.

STZ-induced diabetic rats were intraperitonially injected with 1 mL of 0.1% acetic acid to create Control rats. Five milligrams of PI polyamide to TGF-β1 or Mismatch polyamide dissolved in 1 mL of 0.1% acetic acid was intraperitonially injected twice a week for 3 months, following which the rats were sacrificed and their kidneys were removed.

### 4.8. Measurement of Blood Glucose, HbA1c, Urinary Protein and Renal Function in Rats

Blood glucose was measured in whole blood from rats with a Quick Auto Neo GLU-HK (SHINO-TEST CORPORATION, Tokyo, Japan). HbA1c was measured in whole blood from rats with a RAPIDIA Auto HbA1c-L system (FUJIREBIO Inc, Tokyo, Japan). Serum BUN and creatinine were measured by SRL Inc. (Wako, Saitama, Japan). Collection of 24-hour urine was done in metabolic cages. Urinary albumin and urinary creatinine were measured by SRL Inc. Urinary protein excretion was determined with a Bio-Rad protein assay kit (Bio-Rad Laboratories, Hercules, CA, USA). Urinary albumin excretion was expressed as the ratio of albumin to creatinine.

### 4.9. Immunofluorescence for Podocin

For the immunofluorescence assay, rat kidneys were fixed in 4% paraformaldehyde and cut into 5-μm sections. After the obtained cryosections were deparaffinized with xylene, they were rehydrated with a graded ethanol series. The sections were then blocked with 1% bovine serum albumin (BSA)/PBS for 1 h, incubated in a humidified chamber with anti-podocin (P 0372; Sigma-Aldrich, St Louis, MO, USA) diluted 1:1000 in 1% BSA/PBS for 1 h at room temperature, and then incubated with Alexa Fluor 488 anti-rabbit IgG (A21441; Invitrogen) at room temperature for 1 h. After washing with PBS, the samples were reacted with 4′,6-Diamidino-2-Phenylindole (D9542; Sigma-Aldrich) diluted 1:2000 in pure water, then enclosed in Fluoromount-G (0100-01; Southern Biotech, Birmingham, AL, USA) and observed by fluorescence microscope (IX73; OLYMPUS, Tokyo, Japan).

### 4.10. Western Blot Analysis for Protein Expression in Kidney

Renal medulla from rats was disrupted with lysis buffer (50 mM Tris**·**HCl, pH 8.0, 150 mM NaCl, 0.02% sodium azide, 100 µg/mL phenylmethylsulfonyl fluoride, 1 µg/mL aprotinin, and 1% Triton X-100). Total proteins were extracted and purified with 100 μL of chloroform and 400 μL of methanol. After the protein samples were boiled for 3 min, they were electrophoresed on 8% polyacrylamide gels and then transblotted to nitrocellulose membranes (Bio-Rad Laboratories). The blots were then incubated for 3 h at room temperature with rabbit polyclonal antibody for TGF-β1 (sc-146; Santa Cruz Biotechnology, Santa Cruz, CA, USA) diluted 1:200 and rabbit monoclonal antibody for GAPDH (Sigma-Aldrich) diluted 1:500 as a control in 5% nonfat milk in TBST solution (10 mM Tris**·**HCl, pH 8.0, 150 mM NaCl, and 0.05% Tween 20).

### 4.11. Evaluation of Renal Injury

Three-millimeter-thick paraffin sections of removed renal cortex were stained for semiquantitative evaluation with hematoxylin and eosin or Masson’s trichrome stain. Renal cortical thickness was measured under high magnification (×400). The GIS was obtained with the formula ((0 × n0) + (1 × n1) + (2 × n2) + (3 × n3) + (4 × n4))/50. The tubulointerstitial area was semi-quantified from 20 randomly selected areas of renal cortex. The percentage of involvement of each area showing sclerofibrotic change was estimated and assigned a score of 0, normal; 1, involvement of <10%; 2, involvement of 10% to 30%; 3, involvement of 30% to 50%; or 4, involvement of ≥50% of the area. The TIS was calculated as ((0 × n0) + (1 × n1) + (2 × n2) + (3 × n3) + (4 × n4))/20.

### 4.12. Electron Microscopy of Podocytes

The ultrastructure of the podocytes was assessed with a transmission electron microscope (JEM-1200EX, JEOL, Tokyo, Japan). Before observation, the specimens were fixed in a 2.5% glutaraldehyde solution overnight. After rinsing in PBS, they were fixed for 2 h in a 1% solution of osmium tetroxide (Gibco, Tokyo, Japan). After dehydration with graded acetone, the specimens were soaked in embedding medium. An ultramicrotome (ULTRACUT UCT, Leica, Wien, Austria) was used to create ultra-thin slides that were then subjected to transmission electron microscopy.

### 4.13. Statistical Analyses

Values are reported as means ±SE. The Student *t*-test was used to compare unpaired data, and two-way ANOVA with the Bonferroni/Dunn procedure as a post test was also used. A *p*-value <0.05 was considered to indicate statistical significance.

## 5. Conclusions

We found mesangial proliferation with synthetic phenotype change and podocyte injury, which were improved with the administration of TGF-β1 PI polyamide in the present experiments. These findings suggest that TGF-β1 is a pivotal factor in the progression of diabetic nephropathy, and TGF-β1 PI polyamide may be a feasible as a practical medicine to improve nephropathy.

## Figures and Tables

**Figure 1 molecules-25-00950-f001:**
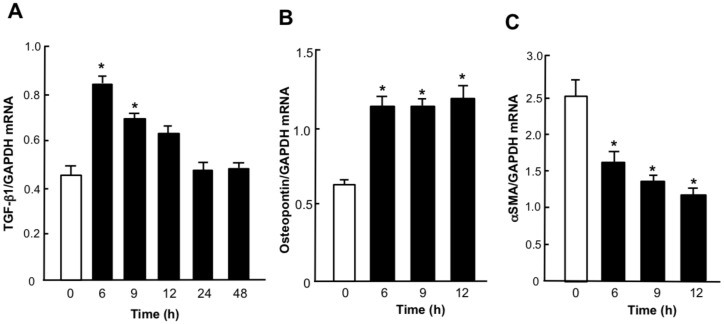
Changes in the expression of TGF-β1 and phenotype marker mRNAs in rat mesangial cells (MCs) with high glucose. Quiescent MCs from Wistar Kyoto rats were incubated with 25 mM glucose for 6 to 48 h. Total RNA was extracted, and the abundance of TGF-β1 (**A**), osteopontin (**B**) and αSMA (**C**) mRNAs was evaluated by real-time polymerase chain reaction analysis. Relative gene expression was analyzed by the comparison to GAPDH mRNA. Data are the mean ± SEM (*n* = 6). * *p* < 0.05 versus 0 h.

**Figure 2 molecules-25-00950-f002:**
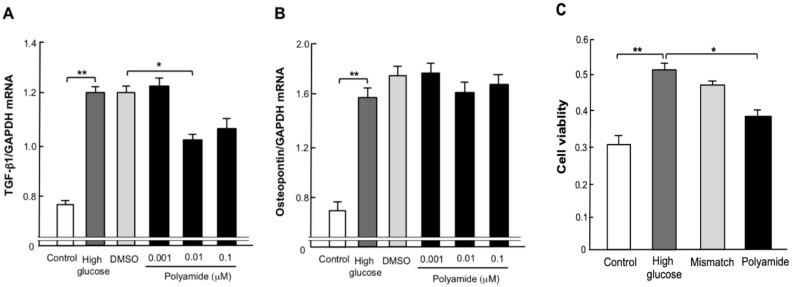
Effects of pyrrole-imidazole (PI) polyamide targeting TGF-β1 on the expression of TGF-β1 and osteopontin mRNAs, and growth of rat mesangial cells (MCs). Quiescent MCs from Wistar Kyoto rats were incubated with 25 mM glucose, 25 mM glucose plus 1 µL/mL dimethyl sulfoxide (DMSO) and 25 mM glucose plus 0.01 to 0.1 µM PI polyamide targeting TGF-β1 for 6 h. Expression of (**A**) TGF-β1 and (**B**) osteopontin mRNAs was evaluated by real-time polymerase chain reaction analysis. Relative gene expression was analyzed by the comparison to GAPDH mRNA. (**C**) MCs were stimulated with 10% FBS in high-glucose DMEM with 0.01 µM PI polyamide targeting TGF-β1 or Mismatch polyamide for 24 h. Growth of MCs was evaluated by WST-1 assay. Data are the mean ± SEM (n=6). * *p* < 0.05, ** *p* < 0.01 between indicated columns.

**Figure 3 molecules-25-00950-f003:**
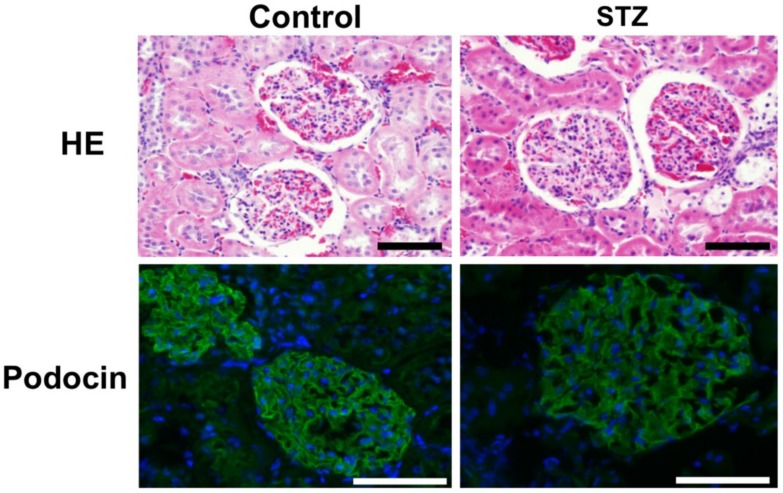
Morphological changes in the glomerulus of diabetic rats at 3 months after injection of streptozotocin (STZ). Diabetes was induced in male Wistar rats by single intraperitoneal injection of 60 mg/kg body weight of STZ. Control rats were intraperitoneally injected with 1 mL of 0.1% acetic acid. Three-millimeter-thick paraffin sections of removed renal cortex were stained with hematoxylin and eosin (HE) or Masson’s trichrome stain. The kidney sections were incubated with anti-podocin and then incubated with Alexa Fluor 488 anti-rabbit IgG. Scale bar indicates 25 µm.

**Figure 4 molecules-25-00950-f004:**
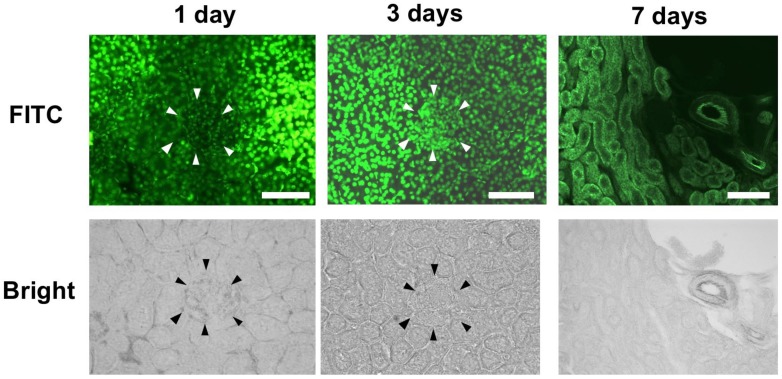
Delivery of pyrrole-imidazole (PI) polyamide targeting TGF-β1 in rat kidney. One milligram of FITC-PI polyamide was injected intraperitoneally into Wistar rats. After 1, 3, and 7 days, the kidneys were removed, and frozen specimens were prepared and viewed. Scale bar indicates 25 µm. The arrow heads distinguish the glomerulus from nephrotubules.

**Figure 5 molecules-25-00950-f005:**
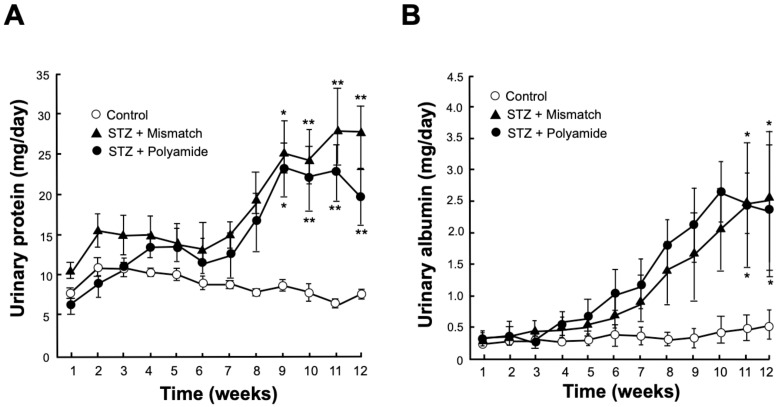
Effects of pyrrole-imidazole (PI) polyamide targeting TGF-β1 or Mismatch PI polyamide on urinary excretions of protein and albumin in streptozotocin (STZ) diabetic rats. Diabetes was induced in male Wistar rats by single intraperitoneal injection of 60 mg/kg body weight of STZ. Control rats were intraperitoneally injected with 1 mL of 0.1% acetic acid. Urinary protein (**A**) and urinary albumin (**B**) were measured in 24-h urine collected in metabolic cages for 12 weeks. Five milligrams of PI polyamide targeting TGF-β1 dissolved in 1 mL of 0.1% acetic acid was intraperitoneally injected twice a week. Data are the mean ± SEM (n=5). * *p* < 0.05, ** *p* < 0.01 versus Control.

**Figure 6 molecules-25-00950-f006:**
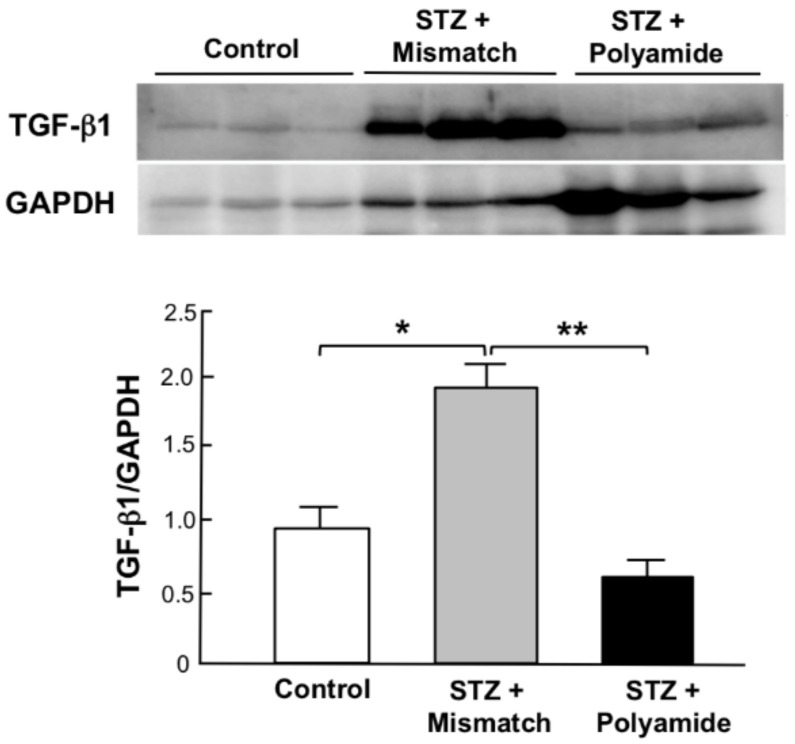
Effects of pyrrole-imidazole (PI) polyamide targeting TGF-β1 on the expression of TGF-β1 protein in kidney from streptozotocin (STZ) diabetic rats. Diabetes was induced in male Wistar rats by single intraperitoneal injection of 60 mg/kg body weight of STZ. Control rats were intraperitoneally injected with 1 mL of 0.1% acetic acid. Five milligrams of PI polyamide targeting TGF-β1 or Mismatch polyamide dissolved in 1 mL of 0.1% acetic acid was intraperitoneally injected twice a week for 3 months, and then the rats were sacrificed to remove their kidneys. Expression of TGF-β1 proteins was assessed by Western blot analysis. Data are the mean ±SEM (*n* = 3). * *p* < 0.05, ** *p* < 0.01 between indicated columns.

**Figure 7 molecules-25-00950-f007:**
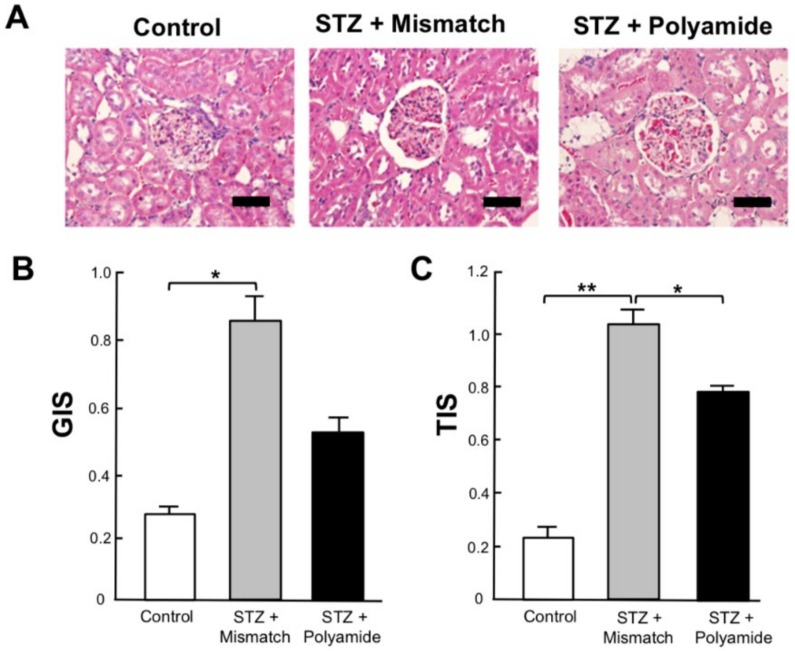
Effects of pyrrole-imidazole (PI) polyamide targeting TGF-β1 on renal injury in streptozotocin (STZ) diabetic rats at 3 months after STZ injection. Diabetes was induced in male Wistar rats by single intraperitoneal injection of 60 mg/kg body weight of STZ. Control rats were intraperitoneally injected with 1 mL of 0.1% acetic acid. Five milligrams of PI polyamide targeting TGF-β1 or Mismatch polyamide dissolved in 1 mL of 0.1% acetic acid was intraperitoneally injected twice a week for 3 months, and then the rats were sacrificed to remove their kidneys. (**A**) The 3-mm paraffin sections of removed renal cortex were stained with hematoxylin and eosin. Scale bar indicates 25 µm. (**B**) Glomerular injury score (GIS). (**C**) Tubulointerstitial injury score (TIS). Data are the mean ± SEM (*n* = 8). * *p* < 0.05, ** *p* < 0.01 between indicated columns.

**Figure 8 molecules-25-00950-f008:**
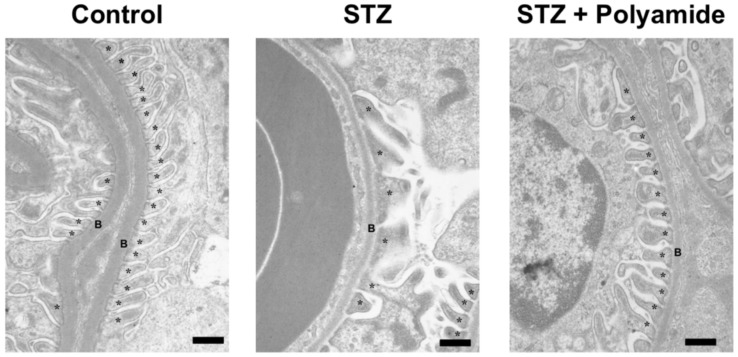
Electron microscopic findings of kidney from control rats, STZ diabetic rats and STZ diabetic rats with pyrrole-imidazole (PI) polyamide targeting TGF-β1. Diabetes was induced in male Wistar rats by single intraperitoneal injection of 60 mg/kg body weight of STZ. Control rats were intraperitoneally injected with 1 mL of 0.1% acetic acid. Five milligrams of PI polyamide targeting TGF-β1 polyamide dissolved in 1 mL of 0.1% acetic acid was intraperitoneally injected twice a week for 3 months, and then the rats were sacrificed to removes their kidneys. The ultrastructure of the podocytes was observed under transmission electron microscopy. Scale bar indicates 1 μm. B–basement membrane, *–podocyte.

**Figure 9 molecules-25-00950-f009:**
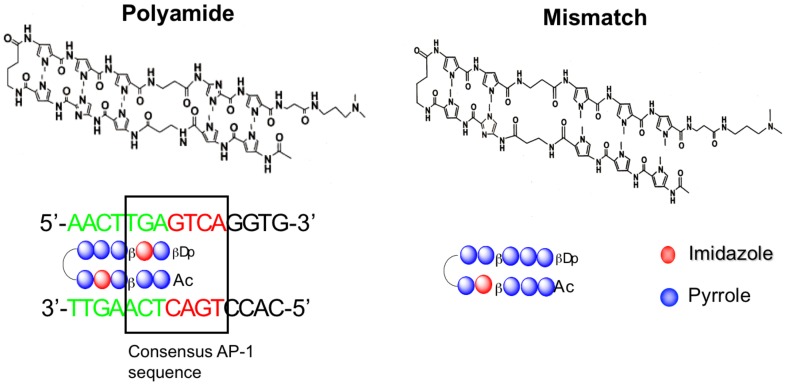
Structure of pyrrole-imidazole (PI) polyamide targeting rat TGF-β1 and mismatch polyamide (Mismatch). PI polyamide targeting TGF-β1 was designed to span the boundary of the activator protein-1 (AP-1) binding site (Box: -2303 to -2297) of the TGF-β1 promoter. Mismatch was designed not to bind transcription binding sites of the promoter. Polyamides were synthesized by solid-phase methods and were purified by HPLC (0.1% AcOH/CH3CN 0 to 50% linear gradient, 0 to 40 min, 254 nm through a Chemcobond 5-ODS-H column).

**Figure 10 molecules-25-00950-f010:**
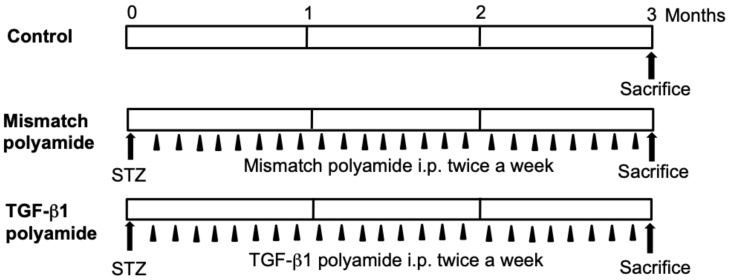
Experimental protocol of the effects of pyrrole-imidazole (PI) polyamide targeting TGF-β1 on renal injury in streptozotocin (STZ) diabetic rats. STZ-induced diabetic rats were intraperitonially injected with 1 mL of 0.1% acetic acid as Control rats. Five milligrams of PI polyamide to TGF-β1 or Mismatch polyamide dissolved in 1 mL of 0.1% acetic acid was intraperitonially injected twice a week for 3 months, and then the rats were sacrificed to remove their kidneys.

**Table 1 molecules-25-00950-t001:** Serological parameters of peripheral blood from Control rats and STZ diabetic rats without or with Mismatch polyamide or PI polyamide targeting TGF-β1.

	Glucose (mg/dL)	HbA1c (%)	BUN (mg/dL)	Cr (mg/dL)
**Control (*n* = 5)**	191.8 ± 2.9	5.3 ± 0.3	24.3 ± 0.5	0.21 ± 0.01
**STZ (*n* = 6)**	510 ± 34	10.8 ± 0.3	22.7 ± 1.3	0.21 ± 0.01
**Mismatch polyamide (*n* = 5)**	537 ± 32	10.7 ± 0.1	21.8 ± 1.6	0.35 ± 0.01
**PI polyamide targeting TGF-β1 (*n* = 6)**	516 ± 0.3	10.5 ± 0.5	25.4 ± 0.3	0.23 ± 0.01

STZ–streptozotocin, PI–pyrrole-imidazole, BUN–blood urea nitrogen, Cr–creatinine. Values are means ± SEM.

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
