# Peer review of "Contribution of TGF-β1 and Effects of Gene Silencer Pyrrole-Imidazole Polyamides Targeting TGF-β1 in Diabetic Nephropathy"

_molecules, 2020, doi:10.3390/molecules25040950_

Round 1

Reviewer 1 Report

The manuscript by Shu Horikoshi et al. (Molecules-702838) is investigating the possibility of treating diabetic nephropathy by inhibiting transforming growth factor (TGF) beta signaling. Peptide compounds pyrrole-imidazole (PI) polyamides that target human TGF-beta1 promoter were used to treat rats with streptozotocin (STZ) induced diabetic nephropathy. Presented results show that PI polyamides decreased expression of TGF-beta1 in the kidney from the STZ-treated rats and partially restore their functionality. 

I have a few comments on the manuscript.

In the legend to Figure 3, the authors describe the treatment of rats with PI polyamides, but the results are not presented.

Most of the experiments were done within 3 months. However, the effect of polyamides on urinary protein and albumin excretion (Fig. 5 C and D) has only been studied for 4 weeks. Why?

In fig. 6 the authors show that treatment with STZ strongly induced expression of TGF-beta1 in kidneys. Was this TGF-beta activated or in the latent form?

In addition, treatment of the rats with polyamides strongly increased level of GAPDH. Authors should use another housekeeping protein for normalization of TGF-beta expression.

Author Response

Manuscript ID: molecules-702838-R1

Reply to Reviewer 1

Comment 1: In the legend to Figure 3, the authors describe the treatment of rats with PI polyamides, but the results are not presented.

Response: Indicated sentences of the treatment of rats with PI polyamides were removed from the legend to Figure 3 in the revised manuscript.

Comment 2: Most of the experiments were done within 3 months. However, the effect of polyamides on urinary protein and albumin excretion (Fig. 5 C and D) has only been studied for 4 weeks. Why?

Response:We already had data of Effects of TGF-β1PI Polyamide on Urinary Excretions of Protein and Albumin in STZ Diabetic Rats for 12 weeks. But in the obtained data TGF-β1PI Polyamide did not significantly decrease urinary excretions of protein and albumin in STZ-diabetic rats. In the revised manuscript, we show the Effects of TGF-β1PI Polyamide on Urinary Excretions of Protein and Albumin in STZ Diabetic Rats for 12 weeks in the revised manuscript as follows.

Urinary excretion of protein were significantly higher in STZ diabetic rats than those in the Control rats at 9 (p< 0.05) , 10, 11 and 12 weeks (p< 0.01) after STZ injection (Figure 5A). Urinary excretions of albumin (p< 0.05) were significantly higher in STZ diabetic rats than those in Control rats at 11 and 12 weeks after STZ injection (Figure 5B). Treatment with TGF-β1 PI polyamide decreased urinary excretions of protein in STZ diabetic rats at 11 and 12 weeks compared to treatment with Mismatch PI polyamide, but statistically not significant (Figure 5A). Treatment with TGF-β1 PI polyamide did not significantly decrease urinary excretion albumin in STZ diabetic rats compared to treatment with Mismatch PI polyamide (Figure 5B).

Comment 3: In fig. 6 the authors show that treatment with STZ strongly induced expression of TGF-beta1 in kidneys. Was this TGF-beta activated or in the latent form?

Response:Protein samples include whole TGF-b1 including the active and latent forms.

Comment 4: In addition, treatment of the rats with polyamides strongly increased level of GAPDH. Authors should use another housekeeping protein for normalization of TGF-beta expression.

Response: As your appropriate indication, expression of GAPDH as a housekeeping protein was increased in kidney from rats treated with PI polyamide. However, the final values were calculated as ratios of TGF-b1/GAPDH, in which the deviation of ratio was not large. Please consider that final conclusion of “Treatment with TGF-β1 PI polyamide significantly (p< 0.01) decreased the abundance of TGF-β1 protein in kidneys from STZ diabetic rats” is correct in Figure 6.

Thank you

Reviewer 2 Report

TGF-b1 has a substantial role in the pathogenesis of diabetic nephropathy. PI polyamide-based gene silencing is one way specifically to decrease gene expression at the level of promoter. Herein, PI polyamides targeting TGF-b1 had been utilized to evaluate its effect on diabetic nephropathy, in vitro and in vivo. High glucose stimulated mesangial cell growth along with TGF-b1/osteopontin mRNA upregulation and a-SMA mRNA downregulation. PI polyamides targeting TGF-b1 decreased high glucose-induced TGF-b1 mRNA expression and cell growth. In STZ diabetic rats, PI polyamides targeting TGF-b1 decreased urinary excretion of protein, miR-23a expression, and morphological changes.

The same strategy had been applied to several experimental studies by the authors and others. The findings only added the diabetic nephropathy to the list without insightful data.

There are many typographic errors and the writing should be carefully edited. The role of TGF-b1, osteopontin, and a-SMA should be described. Their differential profiles after high glucose treatment should be discussed. In cell viability, the effect of PI polyamides targeting TGF-b1 on normal glucose condition should be included. Why the PI polyamides targeting TGF-b1 failed to alter osteopontin mRNA? How about a-SMA mRNA? In rat studies, the STZ control group should be included for biochemical measurement. What is the role of miR-23a? In Discussion, the authors had made a substantial description. However, the statement was not linking to the findings of this manuscript. The parameters were not measured. A substantial mechanistic study is essential to this manuscript.

Author Response

Manuscript ID: molecules-702838

Reply toReviewer 2

Comment 1: There are many typographic errors and the writing should be carefully edited.

Response: We checked carefully again thetypographic errors and the writing. Thank you.

Comment 2: The role of TGF-b1, osteopontin, and a-SMA should be described.

Response: In Discussion, roles of TGF-β1 and osteopontin in the diabetic nephrpathy are descrived as follows “Increases in osteopontin, one of the synthetic phenotype markers and an inflammatory cytokine, strongly correlate with urinary albumin excretion and glomerulosclerosis in diabetic nephropathy [21]. The increased TGF-β1 contributes to the formation of glomerular ECM and albuminemia seen with podocyte injury. In addition, increased USF1 in diabetes mellitus also binds to the promoter of osteopontin, which induces mesangial proliferation that leads to glomerular hypertrophy.”

Could you check in the revised manuscript?

We added aSMA as the contractile phenotype marker in Discussion in the revised manuscript.

Comment 3: Their differential profiles after high glucose treatment should be discussed.

Response:In Discussion, differential profiles after high glucose treatmentare descrived as follows “Exposure to high glucose and fatty acid decreases AMP-kinase that induces the activation of NFkB and the translocation of USF1 into the nucleus, whichstimulates the transcription of TGF-β1 in the hyperglycemic condition [20].”

Could you check in the revised manuscript?

Comment 4: In cell viability, the effect of PI polyamides targeting TGF-b1 on normal glucose condition should be included.

Response: Thank you your appropriate indication.  

Comment 5: Why the PI polyamides targeting TGF-b1 failed to alter osteopontin mRNA? How about a- SMA mRNA?

Response:High glucose condition act to translocate the transcription factor USF1 into nuclei and activate promoter of TGF-β1 andosteopontin. TGF-β1 PI polyamide inhibits the TGF-β1 promoter but not affect osteopontin promoter. The increased TGF-β1 with high glucose changes the phenotype of mesangial cells to the synthetic, by which expression of a-SMA as a contractile phenotype marker was decreased. Thank you your understanding.

Comment 6: In rat studies, the STZ control group should be included for biochemical measurement.

Response: We setted up the Mismatch PI polyamide group that practically be the STZ control group. 

Comment 7: What is the role of miR-23a?

Response:The miR-23a is important microRNA to inhibit the TGF-β1-induced EnDMT in diabetes mellitus. In the present expeiments, the increases in expression of miR-23a mRNA is theoretically strange, which may depend on extraction of RNA from whole kidney including not only gromerular endothelium but also mesangial cells and nephrotubular epithelial cells. We removed data of miR-23a and miR-21 from the revised manuscript.

Comment 8:In Discussion, the authors had made a substantial description. However, the statement was not linking to the findings of this manuscript. The parameters were not measured. A substantial mechanistic study is essential to this manuscript.

Response: Thank you your appropriate comments. We shorten the Discussion section that mainly related to the obtained results.

Thank you

Reviewer 3 Report

Horikoshi S. et.al., were evaluated the protective effect of Pyrrole-Imidazole polyamide (PI) targeting TGF-β1 in Streptozotocin induced Diabetic Nephropathy in rats. Authors examined the effects of PI on mesangial cell growth, podocyte injury, Blood glucose, HbA1c, BUN, Creatinine urinary protein and albumin levels etc. Manuscript has interesting observation that, intraperitoneal administration of PI twice a week for three months improved the degenerations and podocyte injury diabetic nephropathy. Overall the authors suggesting that, TGF-β1 is a key factor in the progression of diabetic nephropathy, and TGF-β1 PI polyamide may be a possible medicine to improve it.

Design of the study is appropriate enough to achieve the aim mentioned in manuscript. Current form of article is well-presented but, needs some improvements.

Some inadequacies and suggestions listed below:

Fig 3: Authors are suggested to quantify the images and show the glomerulus enlargement data in a graphical representation. Fig 4: Authors are suggested to present a better image in case of FITC day1, where only the glomerulus is dark and not the surrounding tubules, to support the statement in line 132-133 Fig 5: Suggested to explain the differences in initial (1 week) urinary protein levels in STZ groups (between Fig 5a and 5C). Why the measurements in control and STZ groups till 11 weeks and 4 weeks in treated groups. It is suggested to present the data till 4 week or 11 weeks in all groups for a better comparison. It is suggested to maintain the uniform color for one group in line/bar graphs in all figures (ex: control -no color bar/thin line, STZ-black bar/bold line, PI-grey, mismatch-light grey) for readers ease. Fig 9: It would be good to present the data of Control, STZ, STZ+PI for one miRNA, in the same graph as in Figure7, to avoid the fold changes for STZ (Seems the experiment was done on different times/days?)

Typo errors:

Line 37, Introduction: diabetes mellitus? Line 445, Methods 4.13: TRIzol?

Author Response

Manuscript ID: molecules-702838

Reply toReviewer 3

Comment 1: Fig 3: Authors are suggested to quantify the images and show the glomerulus enlargement data in a graphical representation.

Response: According to reviewer’s comments, we measured diameters of glomerulus in Control rats and STZ diabetic rats and added results as follows in the revised manuscript: The diameter of glomerulus was 42.4 ± 4.3 mm (n=15) in Control rats and and 49.2 ± 4.5 mm (n=15) in STZ diabetic rats. The diameter of glomerulus in STZ diabetic ratswas significantly (p< 0.05) larger than that in Control rats.

Comment 2: Fig 4: Authors are suggested to present a better image in case of FITC day1, where only the glomerulus is dark and not the surrounding tubules, to support the statement in line 132-133

Response: Figure 4 is changed as a more clear imageof FITC day1 in the revised manuscript.

Comment 3: Fig 5: Suggested to explain the differences in initial (1 week) urinary protein levels in STZ groups (between Fig 5a and 5C). Why the measurements in control and STZ groups till 11 weeks and 4 weeks in treated groups. It is suggested to present the data till 4 week or 11 weeks in all groups for a better comparison.

Response:We already had data of Effects of TGF-β1PI Polyamide on Urinary Excretions of Protein and Albumin in STZ Diabetic Rats for 12 weeks. But in the obtained data TGF-β1PI Polyamide did not significantly decrease urinary excretions of protein and albumin in STZ-diabetic rats. In the revised manuscript, we show the Effects of TGF-β1PI Polyamide on Urinary Excretions of Protein and Albumin in STZ Diabetic Rats for 12 weeks in the revised manuscript as follows.

Urinary excretion of protein were significantly higher in STZ diabetic rats than those in the Control rats at 9 (p< 0.05) , 10, 11 and 12 weeks (p< 0.01) after STZ injection (Figure 5A). Urinary excretions of albumin (p< 0.05) were significantly higher in STZ diabetic rats than those in Control rats at 11 and 12 weeks after STZ injection (Figure 5B). Treatment with TGF-β1 PI polyamide decreased urinary excretions of protein in STZ diabetic rats at 11 and 12 weeks compared to treatment with Mismatch PI polyamide, but statistically not significant (Figure 5A). Treatment with TGF-β1 PI polyamide did not significantly decrease urinary excretion albumin in STZ diabetic rats compared to treatment with Mismatch PI polyamide (Figure 5B).

Comment 4: It is suggested to maintain the uniform color for one group in line/bar graphs in all figures (ex: control -no color bar/thin line, STZ- black bar/bold line, PI-grey, mismatch-light grey) for readers ease.

Response: Figure 2C is changed as reviewer’s suggestion.

Comment 5: Fig 9: It would be good to present the data of Control, STZ, STZ+PI for one miRNA, in the same graph as in Figure7, to avoid the fold changes for STZ (Seems the experiment was done on different times/days?)

Response:The miR-23a is important microRNA to inhibit the TGF-β1-induced EnDMT in diabetes mellitus. In the present expeiments, the increases in expression of miR-23a mRNA is theoretically strange, which may depend on extraction of RNA from whole kidney including not only gromerular endothelium but also mesangial cells and nephrotubular epithelial cells. We removed data of miR-23a and miR-21 from the revised manuscript.

Comment 6: Typo errors: Line 37, Introduction: diabetes mellitus? Line 445, Methods 4.13: TRIzol?

Response: Corrected

Thank you

Round 2

Reviewer 2 Report

The second version of manuscript still existed typographic and wording errors. Besides, the requested supporting data were ignored.

The role and importance of synthetic and contractile types of MC?

In cell viability, the effect of PI polyamides targeting TGF-b1 on normal glucose condition should be included.

Why the PI polyamides targeting TGF-b1 failed to alter osteopontin mRNA? How about a- SMA mRNA?

In rat studies, the STZ control group should be included for biochemical measurement.

In Discussion, the authors had made a substantial description. However, the statement was not linking to the findings of this manuscript. The parameters were not measured. A substantial mechanistic study is essential to this manuscript.

Author Response

Manuscript ID: molecules-702838-R2

Reply toReviewer 2

We appreciate appropriate comments for our mamuscript. We carefuly revised the mamuscript according to your comments as follows.

Comment 1: The role and importance of synthetic and contractile types of MC?

Response: We explained the importance of synthetic and contractile types of MCs as follows in the revised manuscript “Mesenchymal cells including renal MCs show a contractile or synthetic phenotype. MCs in normal glomerulusshow the contractile phenotype, which do not proliferate, whereas MCs in abnormal glomerulus by diabetes mellitus or hypertension shows the synthetic phenotype inducing hyperproliferation with increases in organelles to produce many cytokines [21].”.

Comment 2: In cell viability, the effect of PI polyamides targeting TGF-b1 on normal glucose condition should be included.

Response: Figure 2C shows “Growth of MCs” by WST-1 assay. Its unit is indicated by the Cell viability. Since the Growth of MCs in normal glucose condition is normal level, we did not examine effects of PI polyamides targeting TGF-b1 on normal glucose condition. Thank you your understanding.

Comment 3: Why the PI polyamides targeting TGF-b1 failed to alter osteopontin mRNA? How about a-SMA mRNA?

Response: The high glucose condition increases transcription factor USF1 that activates TGF-b1 promoter as well as osteopontin promoter. PI polyamide targeting TGF-b1 just suppresses TGF-b1 promoter, not the osteopontin promoter. Please consider again this point.

Comment 4: In rat studies, the STZ control group should be included for biochemical measurement.

Response: As the STZ control group, the Mismatch PI polyamide in STZ-diabetic rat  group is practical the STZ control group in Table 1, Figure 5, Figure 6 and Figure 7.

Comment 5: In Discussion, the authors had made a substantial description. However, the statement was not linking to the findings of this manuscript. The parameters were not measured. A substantial mechanistic study is essential to this manuscript.

Response: According to this Reviewer’s comment, we delieted statements not linking to the obtained results from the Discussion section in the revised manuscript.

Please consider again this manuscript for the publication. Thank you.

Reviewer 3 Report

Although most of the points raised in my previous review been addressed satisfactorily but, the paper still needs some minor corrections and clarifications. I would ask the authors to replace the Figure 2 (FITC day1). Current is not acceptable. Increasing the brightness does not support the statement (.. PI polyamide was distributed into the nucleus of the nephron tubule, but not in the glomerulus…)

Author Response

Manuscript ID: molecules-702838-R2

Reply toReviewer 3

Comment 1: I would ask the authors to replace the Figure 2 (FITC day1). Current is not acceptable. Increasing the brightness does not support the statement (.. PI polyamide was distributed into the nucleus of the nephron tubule, but not in the glomerulus…)

Response: We appreciate your reasonable comment, the brightness was equally increased for 1 day, 3 day and 7 day. We found the distribution can be seen mainly in the nucleus of the nephron tubule, and is slightly distributed the glomerulus, at 1 day after injection.Thus the statement was corrected as follows in the revised manuscript:PI polyamide was mainly distributed into the nucleus of the nephron tubule, but was slightly distributed into the glomerulus, at 1 day after injection.

Thank you
